# NMR for Single Ion Magnets

Lucia Gigli [1,2,3], Silvia Di Grande [1,2] , Enrico Ravera [1,2,3,*] , Giacomo Parigi [1,2,3,*]
and Claudio Luchinat [1,2,3,*]

1    Magnetic Resonance Center (CERM), University of Florence, Via L. Sacconi 6, 50019 Sesto Fiorentino, Italy;
     gigli@cerm.unifi.it (L.G.); silvia.digrande@stud.unifi.it (S.D.G.)
2    Department of Chemistry "Ugo Schiff", University of Florence, Via Della Lastruccia 3,
     50019 Sesto Fiorentino, Italy
3    Consorzio Interuniversitario Risonanze Magnetiche di Metalloproteine (CIRMMP), Via L. Sacconi 6,
     50019 Sesto Fiorentino, Italy
*    Correspondence: ravera@cerm.unifi.it (E.R.); parigi@cerm.unifi.it (G.P.); luchinat@cerm.unifi.it (C.L.)

**Abstract:** Nuclear Magnetic Resonance is particularly sensitive to the electronic structure of matter and is thus a powerful tool to characterize in-depth the magnetic properties of a system. NMR is indeed increasingly recognized as an ideal tool to add precious structural information for the development of Single Ion Magnets, small complexes that are recently gaining much popularity due to their quantum computing and spintronics applications. In this review, we recall the theoretical principles of paramagnetic NMR, with particular attention to lanthanoids, and we give an overview of the recent advances in this field.

**Keywords:** Single Ion Magnets; NMR; electronic structure; susceptibility; computational methods



## 1. Introduction

Moving from the large to the small scale, the properties of materials undergo significant variations that result in very interesting effects. In the case of magnetism, systems composed of few paramagnetic centers, Single Molecule Magnets (SMMs), or even single paramagnetic ions, Single Ion Magnets (SIMs), are able to preserve their magnetization due to zero-field degeneracies in the electronic structure. The magnetic hysteresis of these systems, collectively known as single-domain magnets, originates from a bistable ground state, where the two minima are separated by a—possibly—high thermal barrier, called blocking temperature $T_B$. The SMM [1] behavior was first observed in polymetallic complexes [2,3], where it is dominated by exchange interactions between the ions. In subsequent research the SIM behavior has been identified in lanthanoid [4,5] and transition metal ions [6,7], where the magnetic bistability is caused by the electronic structure emerging from the ligand field (LF) splitting (in this review, we use the term "lanthanoids" as recommended to the International Union of Pure and Applied Chemistry [8]).

Due to their magnetic hysteresis, SIMs are expected to find large application in quantum computing [9], first of all as memory devices, and in the field of spintronics. Consistent efforts are indeed devoted to the development of tools that allow for the prediction of the electronic structure from first principles [10–14]. Multireference self-consistent field methods, such as Complete Active Space Self-Consistent Field (CASSCF), are proven successful in reproducing *ab initio* the magnetic properties of paramagnetic centers in general, and the results can be further improved by reintroducing dynamic correlation through the application of Complete Active Space 2nd order Perturbation Theory (CASPT2) or n-Electron Valence State 2nd order Perturbation Theory (NEVPT2) methods. Given the computational requirements of Quantum Chemical (QC) methods, simpler models based on effective electrostatics or angular overlap are also widely used [15–19]. In any case, a solid ground of experimentation is needed to prove the efficacy of the design [20]. The

methodologies of choice for the experimental characterization are Superconducting quantum Interference Device (SQUID) and Cantilever Torque Magnetometry (CTM) [21,22], which are carried out at low temperatures in crystalline solids. The individual contributions of the different positions of the molecule in the unit cell can be deconvolved, and then the data are fit to the crystal field parameters. The use of other methodologies has been described recently [23–26].

In spite of the limited number of papers dealing with NMR of single-domain magnets, it is becoming increasingly apparent that NMR is a powerful tool for their fine characterization and rational design: in NMR-spectroscopic terms, Single Ion Magnets are systems with large and axial magnetic susceptibility anisotropies and fast electron relaxation times at room temperature, making them ideal candidates for an NMR investigation (vide infra). At the same time, the high magnetic susceptibility anisotropies of SIMs are expected to make them suited for NMR applications: a large magnetic anisotropy can provide robust experimental data for the determination of structure and dynamics in biological systems [27–30] and improved contrast agents for Magnetic Resonance Imaging [31,32]. In this review, we will focus on the NMR properties of single-domain magnets, in particular, of lanthanoid Single Ion Magnets (Ln-SIMs) [11,33].

## 2. The SIMs Effects on the NMR Spectra

The magnetic properties of SIMs induce in their NMR spectra significant shifts and relaxation effects [34], which represent an invaluable source of structural information [35].

### 2.1. Susceptibility and NMR Observables

2.1.1. Shifts

As it occurs for a macroscopic object, the magnetic susceptibility tensor $\chi$ of a molecule changes the magnetic field in the vicinity of the molecule. The magnetic susceptibility in a paramagnetic system is due to the presence of unpaired electrons and their interactions among themselves and with nuclei, and can be described through the Van Vleck equation:

$$\chi_{kk} = \frac{\mu_0 \mu_B^2}{k_B T} \frac{\sum_i [\langle \psi_i | L_k + g_e S_k | \psi_i \rangle \langle \psi_i | L_k + g_e S_k | \psi_i \rangle - 2k_B T \sum_{j \neq i} \frac{\langle \psi_i | L_k + g_e S_k | \psi_j \rangle \langle \psi_j | L_k + g_e S_k | \psi_i \rangle}{E_i - E_j}] e^{-\frac{E_i}{k_B T}}}{\sum_i e^{-\frac{E_i}{k_B T}}} \tag{1}$$

where $L_k$ and $S_k$ are the components of the angular momentum and of the spin operator, respectively, $\mu_0$ is the magnetic permeability of a vacuum, $\mu_B$ is the Bohr magneton, $k_B$ is the Boltzmann constant, and $g_e$ is the electron g-factor. The symbols $T$, $E$, and $\psi$ carry the usual meanings of thermodynamic temperature, energy, and quantum state of the system, respectively.

Since most SIMs are based on lanthanoid ions, it is important to recall that, because of the strong spin–orbit coupling, only the total magnetic moment $J$ is defined, thus the Van Vleck equation (Equation (1)) needs to be modified to yield:

$$\chi_{kk} = \frac{\mu_0 \mu_B^2 g_J^2}{k_B T} \frac{\sum_i [\langle \psi_i | J_k | \psi_i \rangle \langle \psi_i | J_k | \psi_i \rangle - 2k_B T \sum_{j \neq i} \frac{\langle \psi_i | J_k | \psi_j \rangle \langle \psi_j | J_k | \psi_i \rangle}{E_i - E_j}] e^{-\frac{E_i}{k_B T}}}{\sum_i e^{-\frac{E_i}{k_B T}}} \tag{2}$$

where $J_k$ are the components of the total angular momentum operator, and the electronic g-factor is given in terms of the spin and the orbital quantum numbers and of their combination to yield $g_J = 1 + \frac{J(J+1) - L(L+1) + S(S+1)}{2J(J+1)}$ [35,36].

The LF causes the energy splitting of the $2J + 1$ sublevels in the spin–orbit–coupled $J$ ground state and can be quantitatively described by the LF parameters. The magnetic susceptibility can thus be calculated from the LF parameters [16,37,38], from the angular overlap [15], or with *ab initio* methods [10,39–41]. As mentioned in the introduction, the calculations can thus be used to provide a physical model for the interpretation of the experimental data.

In NMR, the magnetic susceptibility leaves its marks on several observables: mainly in pseudocontact shifts (PCSs) and residual dipolar couplings (RDCs), but also on relaxation and in other field-dependent effects.

The nuclear spin Hamiltonian contains a term that is proportional to $\chi$ [42–44]. This term is usually represented, according to Kurland and McGarvey [42], as the dipolar interaction between the magnetic moment of the nucleus $\mu_I = \hbar\gamma_I I$ and the average induced electron magnetic moment $\langle \boldsymbol{\mu}_{el} \rangle = \frac{\chi \cdot \boldsymbol{B}_0}{\mu_0}$:

$$H = -\frac{\mu_0}{4\pi}\left[\frac{3(\hbar\gamma_I I_k \boldsymbol{k} \cdot \boldsymbol{r})(\langle \boldsymbol{\mu}_{el}\rangle \cdot \boldsymbol{r})}{r^5} - \frac{\hbar\gamma_I \boldsymbol{k} \cdot \langle \boldsymbol{\mu}_{el}\rangle}{r^3}\right] = -\frac{\hbar\gamma_I B_0}{4\pi r^5} I_k \boldsymbol{k} \cdot \left[3\boldsymbol{r}(\boldsymbol{r}\cdot\boldsymbol{\chi}) - r^2\boldsymbol{\chi}\right]\boldsymbol{k} \quad (3)$$

where the nuclear spin operator $\boldsymbol{I}$ is quantized along the direction $\boldsymbol{k}$ of $\boldsymbol{B}_0$ ($\boldsymbol{I} = I_k\boldsymbol{k}$), $\gamma_I$ is the nuclear gyromagnetic ratio, and $\boldsymbol{r}$ is the metal–nucleus distance. This relation was obtained following a semiempirical treatment [36,45], and has been only recently validated by a complete quantum chemical treatment [46]. The dipolar interaction has an orientational dependence and can be described by a rank-2 tensorial quantity, which is called shielding tensor:

$$\sigma_{PC} = -\frac{1}{4\pi r^5}\left[3\boldsymbol{r}(\boldsymbol{r}\cdot\boldsymbol{\chi}) - r^2\boldsymbol{\chi}\right]. \quad (4)$$

Different orientations of the nucleus–metal vector with respect to the magnetic field direction thus result in different values of shift and, if interconversion between different orientations is impeded, such as in a solid, all different shift values of all nuclei will contribute to the spectrum, generating the so-called powder pattern (Figure 1).

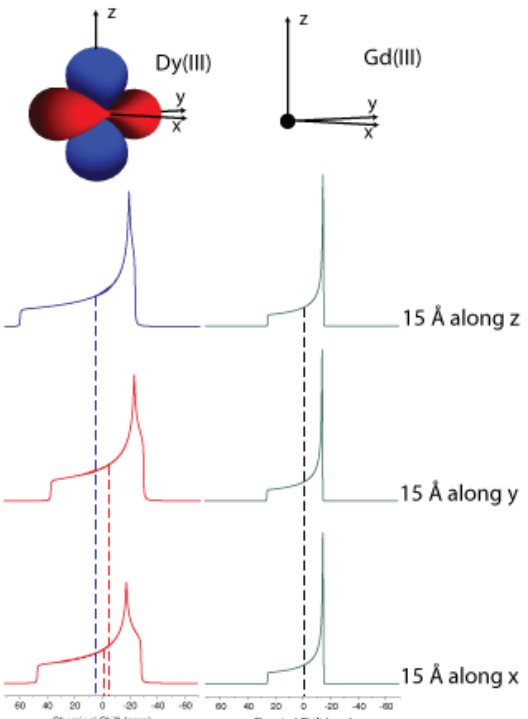

**Figure 1.** Powder patterns for nuclei located along the three principal axes of the magnetic susceptibility tensor. For a rhombic susceptibility tensor (left columns), the three powder patterns are different, and their averages are different from zero (dashed lines). For a spherically symmetric susceptibility (e.g., for a gadolinium(III) ion in a highly symmetric environment), the three powder patterns are identical, and their average equals zero (right column). Therefore, nuclei experience pseudocontact shifts, which have different values for different metal–nucleus positions. Reproduced from ref. [47] with permission from the Royal Society of Chemistry.

On the contrary, in solution the molecule will be free to reorient (see below), and the effect of the shielding will be averaged to its trace. If $\chi$ is isotropic, as in the case of a nondegenerate ground state of high symmetry (e.g., in manganese(II) or gadolinium(III) in $O_h$ or $T_d$ symmetry), no shift will be observed. If, on the contrary, $\chi$ is anisotropic, the average electron magnetic moment $\langle \mu_{el} \rangle$ is not necessarily oriented along the magnetic field (Figure 2), the powder patterns of the different nuclei are different from one another even if at the same distance from the paramagnetic metal, depending on the position of the nucleus in the frame of $\chi$ (Figure 1, left panel), and the reorientational averages are in general different from zero.

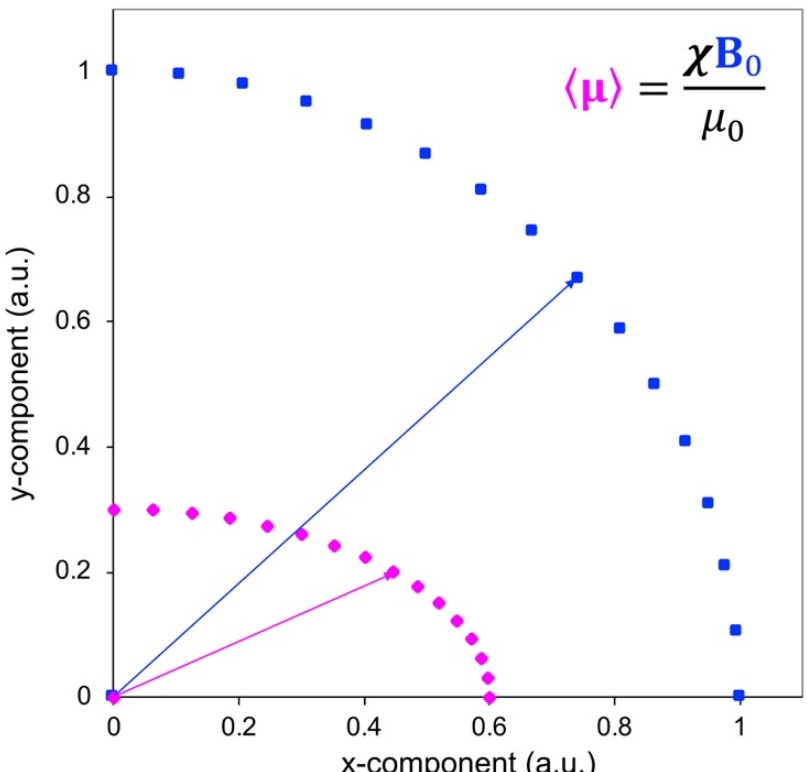

**Figure 2.** Pictorial visualization of the effect of an anisotropic magnetic susceptibility tensor on the average magnetic moment of the electron (magenta), based on the direction of the applied magnetic field (blue).

The average shift values are called pseudocontact shifts and are described by the following equation:

$$\delta_{PCS} = -\frac{1}{3} Tr(\sigma_{PC}) = \frac{1}{12\pi r^3} \left[ \left( \chi_{zz} - \frac{\chi_{xx} + \chi_{yy}}{2} \right) \left( 3\cos^2 \vartheta - 1 \right) + \frac{3}{2} \left( \chi_{yy} - \chi_{yy} \right) \sin^2 \vartheta \cos 2\varphi \right], \tag{5}$$

where $Tr$ denotes the trace of the matrix, $\vartheta$ and $\varphi$ are the spherical angles describing the orientation of $\mathbf{r}$ in the principal frame of the $\chi$ tensor, and $\chi_{xx}$, $\chi_{yy}$ and $\chi_{zz}$ are the principal components of the tensor.

The paramagnetic shift, i.e., the contribution to the NMR shift due to the presence of a paramagnetic metal, may also contain a term arising from unpaired electron spin density at the nucleus, which is called the "Fermi Contact" shift (FCS). Analogous to the dipolar shielding, it is possible to define an FC shielding tensor:

$$\sigma_{FC} = -\frac{1}{\mu_0} \frac{A}{\hbar} \frac{1}{\gamma_I \mu_B} \left( \chi g^{-1} \right) \tag{6}$$

where $A$ is the hyperfine coupling constant, directly proportional to the unpaired electron density at the nucleus. The FCSs are given by the trace of the FC shielding tensor:

$$\delta_{FCS} = -\frac{1}{3} Tr(\sigma_{FC}) = \frac{1}{\mu_0} \frac{A}{\hbar} \frac{1}{3\gamma_I \mu_B} \left( \frac{\chi_{xx}}{g_{xx}} + \frac{\chi_{yy}}{g_{yy}} + \frac{\chi_{zz}}{g_{zz}} \right). \tag{7}$$

These FCSs can provide a contribution larger than that provided by PCSs, and their separation can result cumbersome [48]. However, the low covalency of the bonding in lanthanoids makes this term practically negligible in most cases, except for relatively short metal–nucleus distances, or in ligands with considerable electron delocalization or for ligands carrying unpaired electrons (e.g., neutral double-decker phthalocyaninato complexes [49]).

### 2.1.2. Field-Dependent Effects

In the above section, we assumed implicitly that the molecule in solution is completely free to reorient. This is not necessarily true, as the strong magnetic anisotropy makes some orientations of the molecule with respect to the field more favorable than others. We can therefore introduce an orientation tensor that describes the probability that the molecule is oriented along the principal axes of the magnetic susceptibility tensor:

$$P_{ii} = \frac{\int_\Omega \cos^2 \alpha_i \exp\left(-\frac{E_i}{kT}\right) \sin \alpha_i d\alpha_i d\beta_i}{\int_\Omega \exp\left(-\frac{E_i}{kT}\right) \sin \alpha_i d\alpha_i d\beta_i} = \frac{\int_\Omega \cos^2 \alpha_i \exp\left[\frac{B_0^2}{2\mu_0 kT}\left(\chi_{ii}^{mol}\cos^2\alpha_i + \chi_{jj}^{mol}\sin^2\alpha_i\cos^2\beta_i + \chi_{zz}^{mol}\sin^2\alpha_i\sin^2\beta_i\right)\right] \sin\alpha_i d\alpha_i d\beta_i}{\int_\Omega \exp\left[\frac{B_0^2}{2\mu_0 kT}\left(\chi_{ii}^{mol}\cos^2\alpha_i + \chi_{jj}^{mol}\sin^2\alpha_i\cos^2\beta_i + \chi_{zz}^{mol}\sin^2\alpha_i\sin^2\beta_i\right)\right] \sin\alpha_i d\alpha_i d\beta_i} \tag{8}$$

where $\alpha_i$ and $\beta_i$ are the spherical angles representing the orientation of the magnetic field with respect to the main axes of the magnetic susceptibility tensor. To the first order, the integrals evaluate to [50,51]:

$$P_{ii} = \frac{1}{3}\left[1 + \frac{B_0^2}{5\mu_0 k_B T}\left(\chi_{ii}^{mol} - \chi_{iso}^{mol}\right)\right]. \tag{9}$$

To include the effect of partial orientation on the shifts it is enough to multiply the shielding tensor of Equations (4) and (6) by the orientation tensor $P$ before taking the trace:

$$\delta_{PCS}^{ori} = -Tr(\sigma_{PC} \cdot P) \tag{10}$$

and, analogously:

$$\delta_{FCS}^{ori} = -Tr(\sigma_{FC} \cdot P). \tag{11}$$

The complete forms of the equations are given elsewhere [36]. The effect of the orientation on the observed shifts is negligible at low fields, but it can reach up to a few percentage points at higher fields (Figure 3).

There are other effects that become apparent because of partial molecular alignment: the dipolar couplings between different nuclei and the quadrupolar couplings in nuclei with a nuclear spin quantum number higher than $\frac{1}{2}$ both vanish upon free reorientation because they are represented through traceless tensors. However, in the presence of anisotropic reorientation, the nucleus–nucleus dipolar coupling gives rise to a splitting given by:

$$\Delta\nu_D = \frac{\mu_0 \hbar \gamma_1 \gamma_2}{8\pi^2 r_{12}^5} Tr\left[\begin{pmatrix} 3x^2 - r^2 & 3xy & 3xz \\ 3xy & 3y - r^2 & 3yz \\ 3xz & 3yz & 3z - r^2 \end{pmatrix}\begin{pmatrix} P_{xx} & 0 & 0 \\ 0 & P_{yy} & 0 \\ 0 & 0 & P_{zz} \end{pmatrix}\right] \tag{12}$$

where $r_{12}$ is the distance between the two nuclei and $x$, $y$ and $z$ are the differences between the coordinates of the two nuclei in the principal frame of the tensor $P$. This splitting is called residual dipolar coupling (RDC). If reorientation is anisotropic due to the anisotropy

of the tensor $\chi$, and thus Equation (9) holds, the residual dipolar couplings thus have a paramagnetic origin.

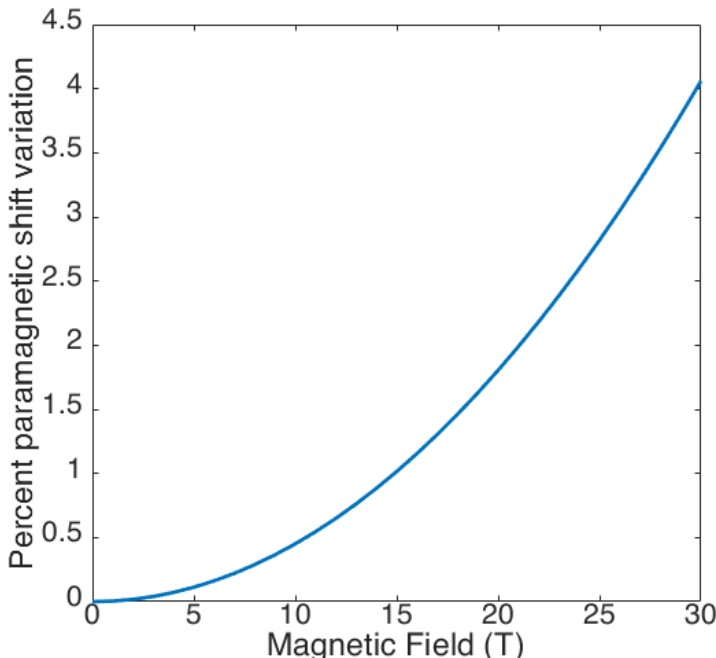

**Figure 3.** Percent variation of the paramagnetic shifts (both PCS and FCS) as a result of field-induced orientation, calculated using the magnetic susceptibility anisotropy reported for the archetypical SIM terbium(III) phthalocyaninato [52].

The quadrupolar coupling depends on the nature of the investigated quadrupolar nuclei, through the quadrupole moment $Q$. The expression for the splitting (residual quadrupolar coupling, RQC) of a transition between two nuclear spin states with $\Delta m_I = \pm 1$ is given by:

$$\Delta \nu_Q = \frac{3eQ}{8I(2I-1)} Tr \left[ \begin{pmatrix} V_{xx} & V_{xy} & V_{xz} \\ V_{yx} & V_{yy} & V_{yz} \\ V_{zx} & V_{zy} & V_{zz} \end{pmatrix} \begin{pmatrix} P_{xx} & 0 & 0 \\ 0 & P_{yy} & 0 \\ 0 & 0 & P_{zz} \end{pmatrix} \right] \tag{13}$$

where $V$ is the symmetric, traceless electric field gradient tensor at the nucleus that, in its principal axes frame, is given by $V = eq \begin{pmatrix} \eta - 1 & 0 & 0 \\ 0 & -\eta - 1 & 0 \\ 0 & 0 & 2 \end{pmatrix}$ [50,53,54]. In most practical cases, the only nucleus with a quadrupole moment sufficiently small so as to be observable by solution NMR is deuterium.

### 2.2. Relaxation

2.2.1. Electron Spin Relaxation

In the environment of the paramagnetic center of interest, molecular motions of the "lattice" and/or of the neighboring spins can generate fluctuating magnetic fields that drive electron spin relaxation. For the sake of simplicity, we will focus our attention on diluted systems, where inter-spin relaxation is reduced or abolished. Under these conditions, the mechanisms through which an electron spin can relax [2,55] are:

- coupling to a phonon that matches the energy difference between the two spin states (direct mechanism);
- coupling to two phonons through an excited state lying within the phonon continuum (Orbach mechanism);

- coupling to two phonons through virtual levels lying inside (Van Vleck or 1st order Raman) or outside of the phonon continuum (2nd order Raman).

These mechanisms are depicted in Figure 3. For lanthanoid ions, low-lying states are available due to the $m_J$ manifold splitting caused by the ligand field. Low-lying states with a strong admixing of electronic and magnetic degrees of freedom are necessary for the two latter mechanisms, which are far more efficient because of the higher density of the acoustic phonons at frequencies higher than the Zeeman splitting (cyan shading in Figure 4) [56]. Therefore, at temperatures higher than $T_B$, relaxation times of the order of picoseconds or lower can be attained. An approximate form of the phonon density is given by the Debye model, with phonons increasing as a cubic power of the frequency but finer representation of phonons and their coupling to spins is becoming possible [56,57].

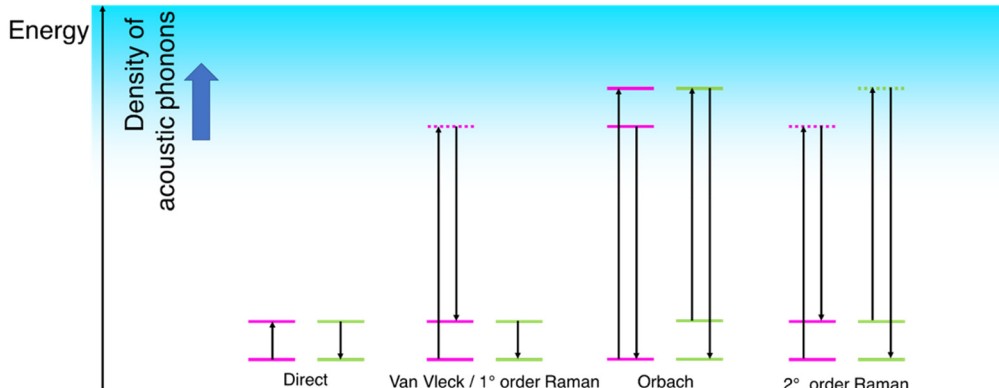

**Figure 4.** Summary of the mechanisms that can cause an electron spin to relax between two levels (depicted as green lines). Magenta lines are energy levels within the lattice. The dashed lines indicate virtual states. The cyan shading indicates the increased density of states of phonons for increased energy in the acoustic range.

As we will see in the following sections, such short electron relaxation times provide relatively inefficient nuclear relaxation, in particular for molecules with relatively short reorientational correlation times and/or at low magnetic fields.

### 2.2.2. Nuclear Spin Relaxation

Nuclear relaxation in paramagnetic systems has a further contribution with respect to diamagnetic systems due to the stochastic modulation of the dipole–dipole interactions between the nuclear magnetic moment $\mu_I$ and the magnetic moment of the unpaired electron(s) $\mu_{el}$. To account for the different strength of the coupling between orbital and spin angular moment, $\mu_{el}$ is written as $\mu_{el} = -\mu_B g \cdot S$ for transition metal ions [36] and $\mu_{el} = -\mu_B g_J J$ for lanthanoid ions (see Table 1) [58].

Two terms can be used to explain the paramagnetic relaxation enhancement, dividing the dipolar interaction of the nuclear magnetic moment with the electron magnetic moment in two terms: (i) the interaction with the zero-averaging component of $\mu_{el}$ (Solomon term, Equations (14) and (15)) and (ii) the interaction with the non-null thermal average of $\mu_{el}$ [35,59,60] (Curie-spin term, Equations (16) and (17)). For the Solomon term, the fluctuations of the dipole–dipole energy can be due to molecular reorientation, chemical exchange, electron relaxation, or a combination thereof. Therefore, the timescale of the fluctuations is defined by a correlation time $\tau_c^{-1} = \tau_R^{-1} + \tau_M^{-1} + R_{1e}$, where $\tau_R$ is the reorientation time, $\tau_M$ the residence time of the nucleus in the proximity of the paramagnetic center, and $R_{1e}$ the electron relaxation rate. Depending on the paramagnetic metal, on the molecular size, and on the residence time of the nucleus, each of these contributions can pass from dictating the value of the correlation time $\tau_c$ to being irrelevant. Because of the very fast electron relaxation typical of all paramagnetic lanthanoid ions except gadolinium(III), in

these ions $\tau_c$ tends to coincide with the electron relaxation time. The Solomon longitudinal and transverse nuclear relaxation rates are described by the following equations:

$$R_{1M}^{\text{Solomon}} = \frac{2}{15}\left(\frac{\mu_0}{4\pi}\right)^2 \frac{\gamma_I^2 g_{\text{iso}}^2 \mu_B^2 S(S+1)}{r^6}\left(\frac{7\tau_{c,2}}{1+\omega_s^2\tau_c^2} + \frac{3\tau_{c,1}}{1+\omega_I^2\tau_c^2}\right), \tag{14}$$

$$R_{2M}^{\text{Solomon}} = \frac{1}{15}\left(\frac{\mu_0}{4\pi}\right)^2 \frac{\gamma_I^2 g_{\text{iso}}^2 \mu_B^2 S(S+1)}{r^6}\left(4\tau_{c,1} + \frac{13\tau_{c,2}}{1+\omega_s^2\tau_c^2} + \frac{3\tau_{c,1}}{1+\omega_I^2\tau_c^2}\right) \tag{15}$$

where $\omega_s = -\gamma_e B_0$ and $\omega_I = -\gamma_I B_0$ are the electron and nuclear Larmor frequency, respectively. In the case of lanthanoid ions other than gadolinium(III), the $J$ quantum number substitutes the $S$ quantum number and $g_J$ substitutes $g_{\text{iso}}$ in the equations above.

**Table 1.** J quantum number and Landé g-factor for lanthanoid ions.

| Ion | J | $g_J$ |
|:---:|:---:|:---:|
| $Ce^{3+}$ | 5/2 | 6/7 |
| $Pr^{3+}$ | 4 | 4/5 |
| $Nd^{3+}$ | 9/2 | 8/11 |
| $Pm^{3+}$ | 4 | 3/5 |
| $Sm^{3+}$ | 5/2 | 2/7 |
| $Eu^{3+}(Sm^{2+})$ | 0 | - |
| $Gd^{3+}(Eu^{2+})$ | 7/2 | 2 |
| $Tb^{3+}$ | 6 | 3/2 |
| $Dy^{3+}$ | 15/2 | 4/3 |
| $Ho^{3+}$ | 8 | 5/4 |
| $Er^{3+}$ | 15/2 | 6/5 |
| $Tm^{3+}$ | 6 | 7/6 |
| $Yb^{3+}$ | 7/2 | 8/7 |

Conversely, the correlation time for Curie spin relaxation is only determined by the reorientation and the nuclear residence time, $\tau_{Curie}^{-1} = \tau_R^{-1} + \tau_M^{-1}$, and not by the electron relaxation, because the thermal average of $\boldsymbol{\mu}_{el}$ (equal to $\frac{\chi \cdot \boldsymbol{B_0}}{\mu_0}$) is already an average over the electron spin states. The Curie-spin relaxation rates can be written in the form:

$$R_{1M}^{\text{Curie}} = \frac{1}{2}\Lambda_\sigma^2\omega_I^2\frac{\tau_{\text{Curie}}}{1+9\omega_I^2\tau_{\text{Curie}}^2} + \frac{2}{15}\Delta_\sigma^2\omega_I^2\frac{\tau_{\text{Curie}}}{1+\omega_I^2\tau_{\text{Curie}}^2} \sim \frac{2}{5}\left(\frac{1}{4\pi}\right)^2\frac{\omega_I^2\chi_{iso}^2}{r^6}\frac{3\tau_{Curie}}{1+\omega_I^2(\tau_{Curie})^2}, \tag{16}$$

$$R_{1M}^{\text{Curie}} = \frac{1}{4}\Lambda_\sigma^2\omega_I^2\frac{\tau_{\text{Curie}}}{1+9\omega_I^2\tau_{\text{Curie}}^2} + \frac{1}{45}\Delta_\sigma^2\omega_I^2\left(4\tau_{\text{Curie}} + \frac{3\tau_{\text{Curie}}}{1+\omega_I^2\tau_{\text{Curie}}^2}\right) \sim \frac{1}{5}\left(\frac{1}{4\pi}\right)^2\frac{\omega_I^2\chi_{iso}^2}{r^6}\left(4\tau_c + \frac{3\tau_{Curie}}{1+\omega_I^2(\tau_{Curie})^2}\right), \tag{17}$$

with

$$\Lambda_\sigma^2 = \left(\sigma_{xy} - \sigma_{yx}\right)^2 + \left(\sigma_{xz} - \sigma_{zx}\right)^2 + \left(\sigma_{yz} - \sigma_{zy}\right)^2, \tag{18}$$

$$\Delta_\sigma^2 = \sigma_{xx}^2 + \sigma_{yy}^2 + \sigma_{zz}^2 - \sigma_{xx}\sigma_{yy} - \sigma_{xx}\sigma_{zz} - \sigma_{yy}\sigma_{zz} + \frac{3}{4}\left[\left(\sigma_{xy}+\sigma_{yx}\right)^2 + \left(\sigma_{xz}+\sigma_{zx}\right)^2 + \left(\sigma_{yz}+\sigma_{zy}\right)^2\right] \tag{19}$$

where $\sigma_{ij}$ are the components of the nuclear shielding tensor, which contains the term $\sigma_{PC}$ (Equation (4)) due to magnetic susceptibility and, potentially, the term $\sigma_{FC}$ as well (Equation (6)). It is curious to observe that, in the presence of a large anisotropy and high magnetic fields, the term in $\Lambda_\sigma^2$ can, in principle, provide a contribution to R$_1$ larger than that to R$_2$ [61,62], though these effects are expected to be too small to be profitably observed.

In Equations (14) and (15) and in the approximate forms of Equations (16) and (17) it is assumed that the $\chi$ tensor is isotropic. It was shown that considering an anisotropic

*g* tensor causes small changes in the Solomon relaxation rates [63–68]. Since the squared thermal average of $\boldsymbol{\mu}_{el}$, corresponding to the prefactor of Equations (16) and (17)

$$\boldsymbol{\mu}_{el}^2 = \left( \frac{g_{\mathrm{iso}}^2 \mu_B^2 S(S+1)}{3kT} \boldsymbol{B}_0 \right)^2 = \frac{\omega_I^2 \chi_{iso}^2}{\mu_0^2}, \tag{20}$$

is much smaller than $g_{\mathrm{iso}}^2 \mu_B^2 S(S+1)/3$, corresponding to the prefactor of Equations (14) and (15), the Curie spin relaxation provides a contribution to the relaxation rates that is non-negligible with respect to the Solomon term, only when $\tau_{Curie} >> \tau_c$. This happens when both the reorientation time and the nuclear residence time are much longer than the electron relaxation correlation time. This is the case of lanthanoids (except gadolinium), which, as already seen, have electron relaxation times as small as picoseconds or less.

Figure 5 shows the dependence of the longitudinal and transverse paramagnetic relaxation rates on the electron relaxation time for a molecule with a reorientation time of 10 ns at 900 MHz and room temperature. The Solomon and Curie contributions to relaxation are also indicated. The Curie contribution to $R_{1M}$ is almost always negligible, except for high spin quantum numbers and very fast electron relaxation, while it can provide a dominant contribution to $R_{2M}$ even for electron relaxation times as large as 1 ns, depending on the value of $S$ (or $J$) and $g_{iso}$ (or $g_J$). In the figure, the curves for the two lanthanoids (samarium(III) and dysprosium(III)) inducing the shortest and the largest relaxation rates are reported, together with the $S = 7/2$, corresponding to gadolinium(III) ion.

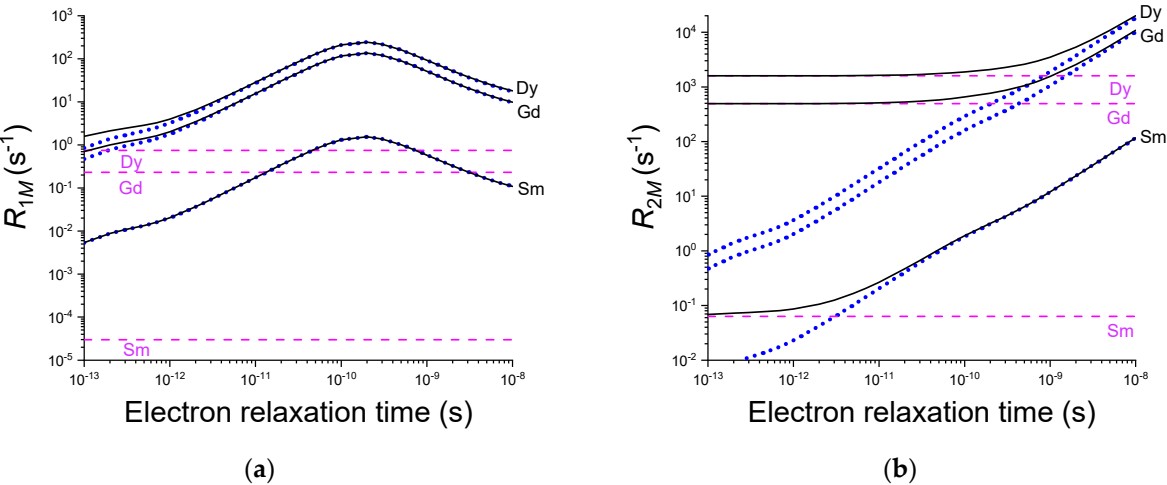

(**a**)

(**b**)

**Figure 5.** Solomon (dotted blue lines) and Curie (dashed pink lines) contributions to nuclear longitudinal (**a**) and transverse (**b**) paramagnetic relaxation (solid black lines) as a function of the electron relaxation time for samarium(III), gadolinium(III) and dysprosium(III). A non-exchangeable $^1$H nucleus was considered at a distance r = 10 Å from the paramagnetic metal, in a molecule with a reorientation time $\tau_R = 10 \; ns$, in a magnetic field of 900 MHz proton Larmor frequency.

Figure 6 shows the relative magnitude of $\frac{1}{15} \left( \frac{\mu_0}{4\pi} \right)^2 \gamma_I^2 g_J^2 \mu_B^2 J(J+1)$ and $\frac{1}{45} \left( \frac{\mu_0}{4\pi} \right)^2 \frac{\gamma_I^2 g_J^4 \mu_B^4 B_0^2 \, J^2(J+1)^2}{k^2 T^2}$ at room temperature, corresponding to the prefactors in the Solomon and Curie spin relaxation equations, respectively. The positions in the plots corresponding to the different lanthanoids are also indicated. The figure clearly shows that (i) the prefactor in the Solomon term is always orders of magnitude larger than the prefactor in the Curie term, and that (ii) the lanthanoids inducing the largest nuclear transverse relaxation rates under the same electron relaxation time and reorientation time are $Dy^{3+}$ and $Ho^{3+}$, followed, in order, by $Tb^{3+}$, $Er^{3+}$, $Gd^{3+}$, $Tm^{3+}$, $Yb^{3+}$, $Ce^{3+}$, and $Sm^{3+}$.

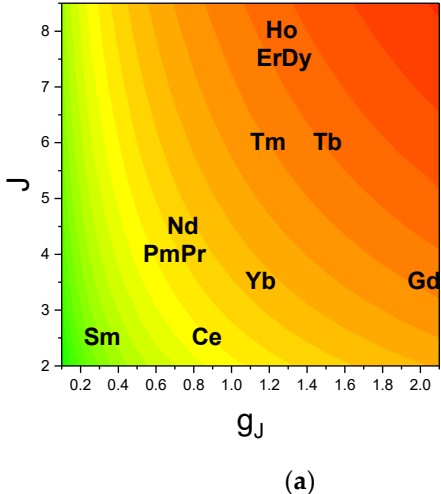

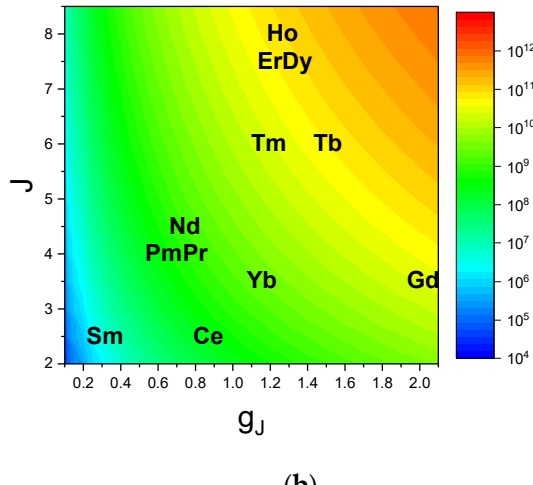

(**a**)

(**b**)

**Figure 6.** Solomon (**a**) and Curie (**b**) dependences on $J$ and $g_J$ of nuclear transverse relaxation. The colors indicate the amplitude of the factors in the equations for $R_{2M}^{Solomon}$ and $R_{2M}^{Curie}$ in front of the parentheses containing the spectral density functions, to be divided by $r^6$, with $r$ in Ångström.

In the case of a non-negligible spin density at the nucleus, contact coupling also occurs and can be described by the Bloembergen equations [69]:

$$R_{1M} = \frac{2}{3}S(S+1)\left(\frac{A}{\hbar}\right)^2\frac{\tau_{c,2}}{1+\omega_s^2\tau_{c,2}^2},\tag{21}$$

$$R_{2M} = \frac{1}{3}S(S+1)\left(\frac{A}{\hbar}\right)^2\left(\tau_{c,1}+\frac{\tau_{c,2}}{1+\omega_s^2\tau_{c,2}^2}\right)\tag{22}$$

where $\tau_{c,1} = R_{1e} + 1/\tau_M$ and $\tau_{c,2} = R_{2e} + 1/\tau_M$.

All equations above assume that the energy of the electron spin states is determined by the Zeeman interaction between the magnetic moment and the applied magnetic field. It was shown that dramatic effects are caused by the inclusion of the zero-field splitting, which may be present in paramagnetic transition metal ions with $S > 1/2$. In fact, the zero-field splitting can largely affect the energy of the electron spin states, especially at low fields. As a result, the differences in energy related to most electronic spin transitions can be much larger than what is calculated from the Zeeman interaction only, thus accounting for changes in the transition probabilities and, in turn, in the nuclear relaxation rates [65,70,71]. In the presence of zero-field splitting, the nuclear relaxation rates not only depend on the distance between nucleus and unpaired electron(s) but also on the angular position of the nucleus with respect to the zero-field splitting tensor axes [36].

In lanthanoids, crystal-field effects remove the degeneracy of the electronic levels at zero magnetic field. This splitting of the electronic levels at zero field is typically modeled by a tensor with the same form of the zero-field splitting tensor. Recently, it was experimentally shown that indeed there is an angular dependence in the relaxation rates measured for a paramagnetic lanthanoid(III) complex [68]; this dependence was modeled using the parametric equation:

$$R_{1M}^{Solomon} = \frac{2}{3}\left(\frac{\mu_0}{4\pi}\right)^2\frac{\gamma_I^2}{r^6}Tr\left[\left(3\hat{r}\hat{r}^T-1\right)^2\boldsymbol{G}(\omega_I)\right]\tag{23}$$

where $Tr$ denotes the trace of the matrix, $\hat{r}$ is the unit vector pointing in the direction of $r$, and the six independent components of the symmetric spectral density tensor $\boldsymbol{G}(\omega_I)$ are treated as fitting parameters.

## 3. NMR for SIMs and Vice Versa

In systems with a degenerate ground state, the magnetic susceptibility anisotropy is usually rather large, and the electron relaxation time short. From what we have seen in the two paragraphs above, this implies that in these cases the paramagnetic shifts of the NMR-active nuclei are rather large and, assuming that the unpaired spin density on the observed nucleus is negligible, they only depend on the magnetic susceptibility anisotropy. At the same time, nuclear relaxation is not particularly severe, especially for small molecules and low magnetic fields. On these grounds, it is not surprising that NMR is a particularly suitable tool for the characterization of the electronic structure of those systems that feature SIM behavior [34,72,73].

As already mentioned, the first systems identified to have SIM behavior were double-decker phthalocyanine complexes with $Tb^{3+}$ or $Dy^{3+}$ [6]. The preparation and characterization of these complexes dates back to the late eighties [74,75]. A complete NMR characterization of double- and triple-decker (either SIMs with a single paramagnetic center or dinuclear SMMs) has followed more recently, due to the difficulties that are related to the detection of the signals of strongly paramagnetic systems as described above [49,52,76–78]. Of note, porphyrin double-deckers were addressed earlier [79,80].

Sugita and co-workers proposed an NMR-based method to determine the LF parameters from a series of isomorphically structured lanthanoid complexes that simultaneously reproduce the $^1H$ NMR paramagnetic shifts and the magnetic susceptibilities measured at various temperatures [81]. An NMR-only method for the LF parameters determination was proposed by the group of Enders [37,49,82,83]. The approach is based on the partial alignment effects, RDCs and RQCs, which depend on the paramagnetism (see paragraph 1.2 above and chapter 3 in ref. [35]) of the system [50,84]. The analysis of hyperfine shifts from $^1H$ and $^2H$ NMR spectra at different temperatures, combined with structural models, allows for the determination of the axial magnetic susceptibility tensors for a lanthanoid ions series. The temperature dependence of the $\chi_{ax} = \chi_{zz} - \frac{\chi_{xx} + \chi_{yy}}{2}$ values is then used to derive the three axial LF parameters (Figure 7) [37].

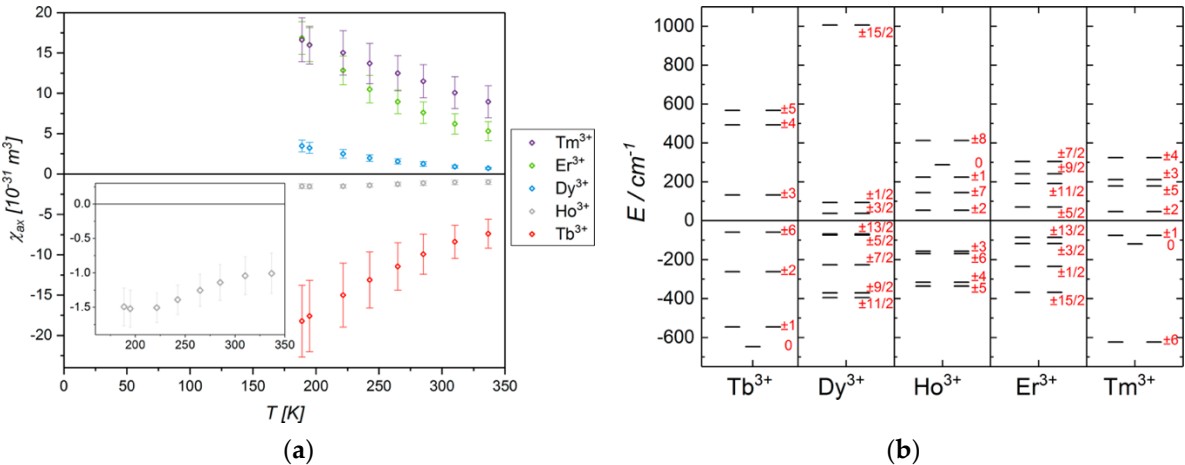

**Figure 7.** Temperature dependence of the magnetic susceptibility anisotropies (**a**), and relative energies of the different $m_J$ sublevels obtained from the fitted Ligand Field parameters (**b**). The investigated compounds are the Ln(tdnCOT)$^{2-}$ series (tdnCOT = (6Z,14Z)-5,8,13,16-tetrahydrocycloocta [1,2-b:5,6-b']-dinaphtalene; the values for the Ho$^{3+}$ analogues are enlarged in the inset). Reprinted with permission from ref. [37]. Copyright 2017 American Chemical Society.

The results of this work also allow for a general consideration about the interplay between theory and experiment, which is necessary, but should be used with caution, because computational methods, although powerful, can result in discrepancies with experimental values. Remarkable is, indeed, the case of the Dy$^{3+}$ ion, for which the ground state results to be the $m_J = |\pm 11/2\rangle$ state through the NMR-based method and the

$m_J = |\pm 9/2\rangle$ state from the *ab initio* calculations, and the $\chi_{ax}$ value is negative according to the computations, positive according to the experiments [37].

As mentioned in the introduction, Ln-SIMs are even ideal candidates as shift-inducing tags [85–87], due to their high magnetic susceptibility anisotropy, tunable paramagnetic properties, and relative stability. Systems for which lanthanoid substitutions are now quite common are proteins, because even though not naturally present in biological systems, they can substitute ions such as $Mg^{2+}$ and $Ca^{2+}$, due to their similar ionic radii [85]. We note the exception of a recently discovered Ln-binding protein [88–91]. Although not preferred because of their significantly lower magnetic susceptibility anisotropy with respect to lanthanoids, transition metal ions also can display Single Ion Magnets behavior [6,92,93]. Intriguingly, in the quest for the achievement of highly efficient SIMs, complexes with large values of the zero-field splitting and large g-anisotropies [94] that correspond to a large anisotropy of the magnetic susceptibility are obtained, as in the case of trigonal prismatic coordinated cobalt(II) cage complexes [95,96]. The large PCSs and the moderate paramagnetic relaxation enhancement exerted by cobalt(II) are nicely highlighted in Figure 8. The $^1$H-NMR signals arise from the long alkyl chain attached to the boron atom. Because the O–B bonds impede the transmission of the spin density, the contact shifts are negligible.

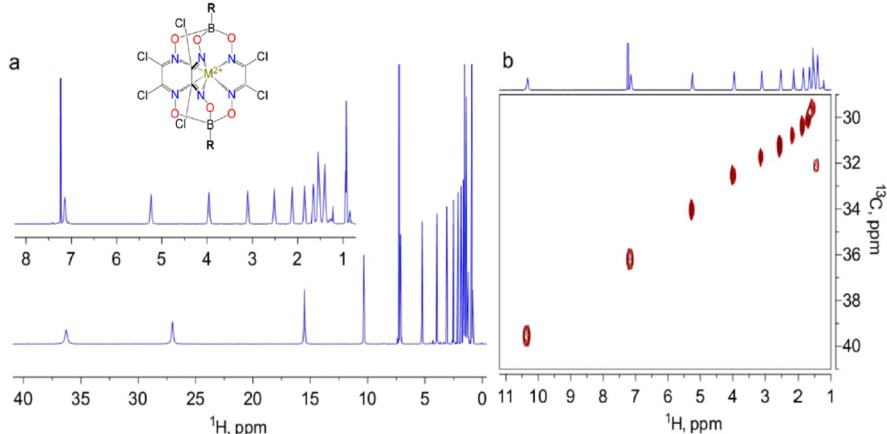

**Figure 8.** $^1$H (**a**) and fragment of $^1$H-$^{13}$C Heteronuclear Single Quantum Coherence (**b**) NMR spectra (600 MHz, 298 K, 5 mM in solution CDCl$_3$) of the Co(II) complex shown in the inset (M$^{2+}$ = Co$^{2+}$, R = *n*-C$_{16}$-H$_{33}$). The sharp signal at 7.25 ppm corresponds to the residual signal of the solvent. Reprinted with permission from ref. [95], copyright 2014 American Chemical Society.

## 4. Solid-State NMR of Paramagnetic Molecules and Possible Effect of Dynamics on Relaxation in the Solid State

As we have already mentioned (Figure 1), in a solid sample all the nuclei (which can have different shielding values) contribute to the observed powder pattern. It is however important to remark that, as long as interconversions between different positions with respect to the magnetic field are impeded, each nucleus does not randomly explore with time the different values of shift in the powder pattern, and therefore its energy is not modulated. Therefore, Curie spin relaxation is, in principle, absent in solids (vide infra), and high resolution spectra can be achieved under Magic Angle Spinning [97]. In the following, we will review some notable applications of solid-state NMR of paramagnetic complexes. In those solids that are characterized by periodicity, i.e., crystals, there are intrinsically strong interactions between each spin and its neighbors. The intermolecular contributions can be comparable to the intramolecular ones, and sizably complicate the spectra. Therefore, they need to be accounted for when fitting the data [98–100]. This complication, however, provides a wealth of additional information: if the lattice parameters are known, it is possible to simultaneously determine the crystal structures and the magnetic susceptibility starting from the experimental values of shifts and the dipolar shielding anisotropy patterns [101]. Because of the short electron relaxation time and the

absence of Curie-spin relaxation, the solid-state spectra of Ln-based compounds are easily observable [102,103], and the fitting of the anisotropy parameters can then be used to obtain the LF parameters [104]. Given that the intensity of each signal is spread out over shift ranges easily spanning over 1000 ppm, the effect of paramagnetic doping is that of reducing the signal intensity of diamagnetic systems. This effect can be used to extract information about the distribution of the paramagnetic centers, and might be relevant for, e.g., the creation of phosphors [105,106]. Further lineshape analysis can be used to support the estimates [107].

But what happens when the ligand field around the metal center is fluxional? There are several reports of systems that undergo an easy-axis to easy-plane transition with increasing temperature, or for which the principal axes of magnetic susceptibility change dramatically in response to minor changes in the coordination environment [108–113]. Variations in the fundamental state configuration cause orientation changes of the axes of the magnetic susceptibility and its values. As an example, the changes in the axiality degree for a $C_3$-symmetric nine-coordinated lanthanoid series [LnL$^1$] (Ln = Dy, Er, Yb and H$_3$L$^1$ = 1,4,7-tris[(6-carboxypyridin-2-yl)methyl]-1,4,7-triazacyclonane) as a function of the second-rank axial crystal field parameter $B_2^0$ are shown in Figure 9 [114].

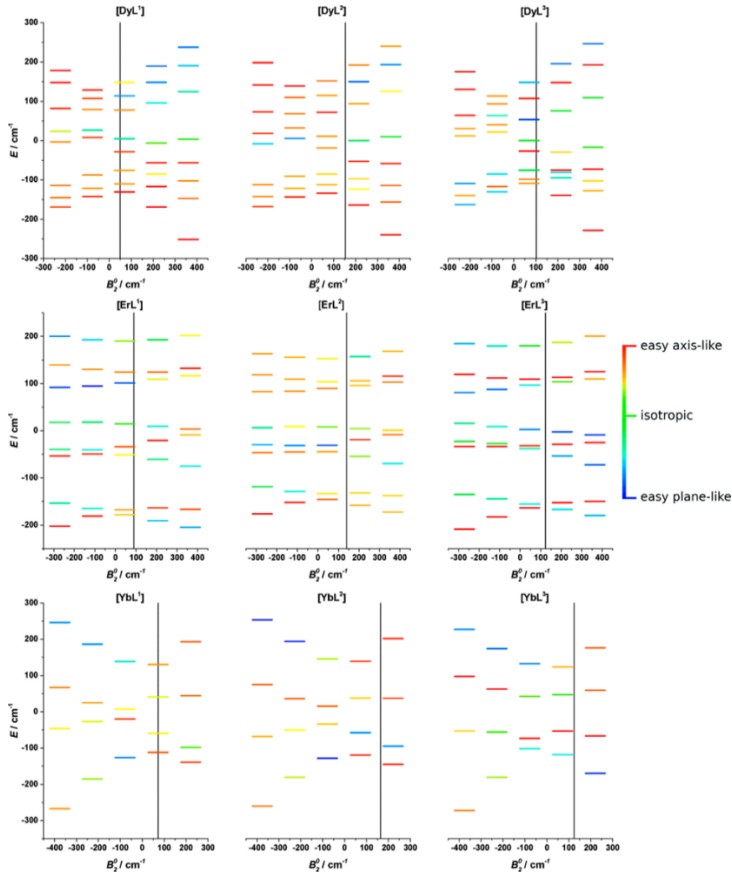

**Figure 9.** Changes in the energies of the ground state *J* multiplets for [DyL$^{1-3}$], [ErL$^{1-3}$], and [YbL$^{1-3}$] in response to the variation of the axial crystal field parameter $B_2^0$. The color code indicates the degree of axiality (toward easy-axis or easy-plane) in the principal g-values anisotropy for each Kramers doublet. Energies are referenced to the barycenter for each parameter set. Vertical lines correspond to the $B_2^0$ value obtained from CASSCF-SO calculations based on XRD data. Reprinted with permission from ref. [114], copyright 2019 American Chemical Society.

These drastic changes in the susceptibility have been shown to have consequences on the magnetic susceptibility as observed by NMR, in terms of changes in the principal axes of the magnetic susceptibility across a homologous series of lanthanoid(III) compounds [115].

Another example along these lines is the change in the magnitude of the anisotropy in $M_3[Yb(BINOL)_3]$ (M = Li, Na, K; BINOL = enantiopure 1,1′-bis(2-naphtol)), which decreases when the radius of the counterion is increased: the anisotropies with sodium(I) and potassium(I) as counterions are 43.8 and 16.0% of the anisotropy with lithium(I) as counterion, respectively (Figure 10) [116].

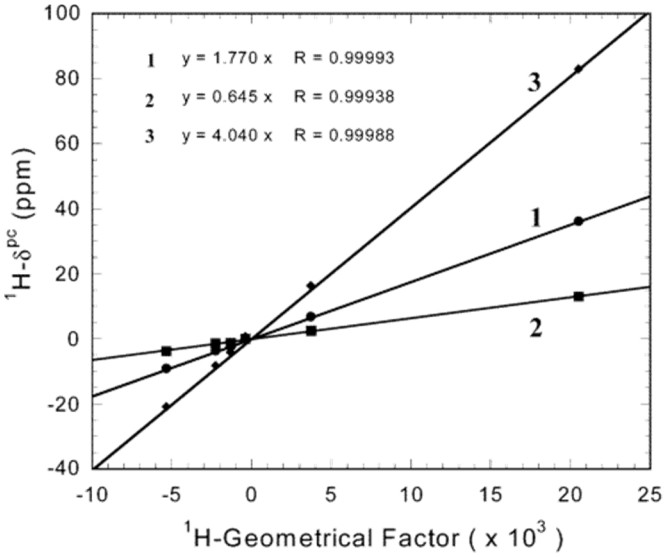

**Figure 10.** Relation between pseudocontact shifts and Geometrical Factors $((3\cos^2\theta - 1)/r^3$ in Equation (5)) for $Na_3[Yb((S)-BINOL)_3]$ (1), $K_3[Yb((S)-BINOL)_3]$ (2), and $Li_3[Yb((S)-BINOL)_3]$ (3). Reproduced with permission from ref. [116], copyright 2003 American Chemical Society.

We proposed earlier that a modulation of the magnetic susceptibility could provide an additional relaxation mechanism in paramagnetic solids (see ref. [47] and chapter 5 in ref. [35]), which can be related to a Curie-spin-like type of relaxation. Here, we would like to point out that the impact of this additional term on the observation of the signals may not be negligible. To provide a quantitative description, we approximated this effect to have the same functional form of Curie-spin relaxation (Equations (16) and (17)). A large part of the shielding anisotropy will not be affected by the averaging of the susceptibility, hence it will only give rise to a powder pattern as described above (Equation (3) and Figure 1). However, there will be a part of the shielding anisotropy given by the difference between the largest shielding values and the averaged values. Depending on the intensity of the applied magnetic field, which increases the frequency range of the shielding anisotropy, and on the rate of modulation of the magnetic susceptibility tensor, this effect can have a sizable impact on the linewidths. In Figure 11 we show the expected effect on the transverse relaxation in the solid state for a system where the main magnetization axis can jump randomly between three perpendicular orientations.

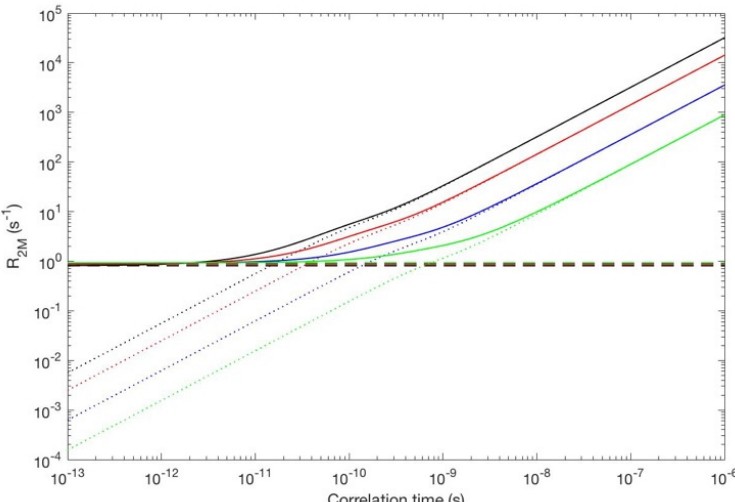

**Figure 11.** Additional Curie-spin-like contribution to the transverse relaxation of a $^1$H nucleus 10 Å away from a dysprosium(III) complex where the magnetic susceptibility (with eigenvalues $5 \times 10^{-30}$, $5 \times 10^{-30}$, and $1 \times 10^{-30}$, respectively) can switch between three orthogonal orientations, with a correlation time ranging from $1 \times 10^{-13}$ to $1 \times 10^{-6}$, evaluated at 28.2 T (black), 18.8 T (red), 9.4 T (blue), 4.7 T (green). The Solomon relaxation evaluated with an electron relaxation time of $1 \times 10^{-13}$ s, at the same fields—which does not depend on the correlation time of the susceptibility reorientation and is, therefore, constant—is also reported.

## 5. Conclusions

In this review we give an overview of the NMR properties of Single Ion Magnets, with a specific focus on lanthanoid complexes. The sizable impact of the lanthanoid ions on shifts and relaxation provides a wealth of information about the system under investigation, as it is demonstrated by the recent literature examples. On the one hand, NMR is a powerful tool for the SIMs characterization, thus allowing for the rational design and application of SIMs in the fields of quantum computing and spintronics. On the other hand, it is apparent that Ln-SIMs are ideal candidates to the generation of paramagnetic tags for structural biology applications and as contrast agents for MRI. Finally, we believe that the examples we examined demonstrate that experimental determination of the magnetic parameters and theoretical models need to be combined: the experimental observations may hide effects of mobility or structural rearrangements that can be only grasped through modeling, and computational data need to be carefully compared to several experimental values to avoid their possible misinterpretation.

**Author Contributions:** This review was conceived by E.R., G.P. and C.L., and was developed, prepared, and written with the contribution of L.G., S.D.G., E.R., G.P. and C.L. All authors have read and agreed to the published version of the manuscript.

**Funding:** This work has been supported by the Fondazione Cassa di Risparmio di Firenze, the Italian Ministero della Salute through the grant GR-2016-02361586, and the Italian Ministero dell'Istruzione, dell'Università e della Ricerca through the grant PRIN 2017A2KEPL and through the "Progetto Dipartimenti di Eccellenza 2018–2022" to the Department of Chemistry "Ugo Schiff" of the University of Florence. E.R. acknowledges the support from the University of Florence through the "Progetti Competitivi per Ricercatori". The authors acknowledge the support and the use of resources of Instruct-ERIC, a landmark ESFRI project, and specifically the CERM/CIRMMP Italy center. This work was also supported by Instruct-ULTRA (Grant 731005), an EU H2020 project to further develop the services of Instruct-ERIC.

**Institutional Review Board Statement:** Not applicable.

**Informed Consent Statement:** Not applicable.

**Data Availability Statement:** Not applicable.

**Acknowledgments:** We wish to dedicate this work to Sandro Bencini. Sandro, who had been a University mate in Chemistry of the senior author, C.L., was a rigorous scientist, with a profound theoretical knowledge, whose untimely passing caused a loss in the School of Inorganic Chemistry in Florence.

**Conflicts of Interest:** The authors declare no conflict of interest.

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
