# Peer review of "NMR for Single Ion Magnets"

_magnetochemistry, doi:10.3390/magnetochemistry7070096_

Round 1
Reviewer 1 Report
In this work Lucia Gigli et al. summarise the theoretical principles of NMR for single ion magnets. It is a comprehensive review that cover the most important aspects of the lanthanoids in paramagnetic NMR and the recent advances within this field.
In my opinion this is a remarkable review, properly organized, structured and redacted that includes the relevant references of the field. Therefore, I recommend its publication with only two minor comments for the authors.
- Some terms such as CASSCF, CASPT2, NEVPT2 or SQUID are introduced in the text without the corresponding definition.
- Figures 1 and 7 present a poor resolution.
Author Response
In this work Lucia Gigli et al. summarise the theoretical principles of NMR for single ion magnets. It is a comprehensive review that cover the most important aspects of the lanthanoids in paramagnetic NMR and the recent advances within this field.
In my opinion this is a remarkable review, properly organized, structured and redacted that includes the relevant references of the field. Therefore, I recommend its publication with only two minor comments for the authors.
We thank the reviewer for the appreciation. We have introduced the changes as suggested
- Some terms such as CASSCF, CASPT2, NEVPT2 or SQUID are introduced in the text without the corresponding definition.
Done
- Figures 1 and 7 present a poor resolution.
We have now provided the figures at the highest available resolution.
Reviewer 2 Report
The review „NMR for Single Ion Magnets” submitted to the journal Magnetochemistry, is written by experts in the field of NMR spectroscopy of paramagnetic molecules. The focus of this article lies on lanthanide based SIMs while touching d-block systems at times. The article covers the theoretical background necessary for understanding the field and gives some examples of NMR studies of SIMs in solution (chapter 3) and in the solid-state (chapter 4). It should be published after a minor revision:
Major points of criticism:
It is clear that polynuclear d-block SMMs like the Mn12 cluster are not covered in this article, however I wonder whether NMR studies of dinuclear Ln-based SMMs should be ignored or not (J. Am. Chem. Soc. 2013, 135, 14349. DOI: 10.1021/ja4069485 or Nature Communications 2019, 10, 571. DOI: 10.1038/s41467-019-08513-6).
On the other hand, the first comprehensive NMR study on a derivative of the archetypal SIM TbPc2 must definitely be mentioned in such a review (Chem.-Eur. J. 2015, 21, 14421. DOI: 10.1002/chem.201501944). Nice NMR studies have also been made with fullerene trapped SIMs (Chemical Science 2015, 6, 2328. DOI: 10.1039/C5SC00154D)
Minor points:
Line 34:
I am not aware if there is already an “application” of SIMs. At least the authors do not give a reference. Therefore, I think it would be better to change
“SIMs find large applications in …..”
to “are expected to find large applications in….”
Line 141:
I agree that for several NMR signals of SIMs FCS is negligible, however, it has nevertheless to be considered for some of the resonances (e.g. 13C atoms and aromatic C-1H in LnPc2 anion). When the ligands carry a delocalized unpaired electron like in the neutral from of LnPc2 FCS has to be considered. Therefore, I suggest changing from:
“.., except for relatively short metal-nucleus distances.”
to
“.., except for relatively short metal-nucleus distances, or in ligands with considerable electron delocalisation or for ligands carrying unpaired electrons (e.g. neutral double decker phthalocyaninato complexes).”
Chapter 2.1.2 Field Dependent effects
I always appreciated very much when estimations of several effects are presented in numbers or in nice diagrams (e.g. in this article Figures 4 and 5 and many examples in the fantastic book “NMR of paramagnetic molecules”). Therefore it would be great to see an estimation of the field dependant effects presented in equations 10 and 11 (e.g. for TbPc2 at a field of 24 Tesla).
Figure 4:
a) These curves may lead to misunderstandings: Curves are given for S=1/2 and S=7/2. On the other hand it is stated, that electron relaxation depends on J for lanthanides. Therefore the curves presented for S=1/2 or S=7/2 may represent d-block ions? However, S=7/2 is not possible for transition metals. Perhaps the diagrams could be changed to S=5/2 (corresponding to high-spin Mn2+).
b) the blue dots are hardly visible (especially in the left diagram).
Line 322:
Reference 69 does not seem to be a SIM paper and is inadequate here
Line 325 - 326:
The statement “… and their NMR characterization has followed shortly after [72,73]” is not correct and needs to be changed. The references given here are not from phthalocyaninato double deckers but from porphyrin double deckers. The NMR characterization of LnPc2-derivatives was done much later.
Line 363:
The fact that FCS is negligible here has to do with an “insulating unit –O-B-”. In order to make this clearer to the readers one might insert a sentence like:
“The signals arise from the long alkyl chain attached to the boron atom. As spin density is hardly transmitted through the O-B-unit contact shifts are negligible.”
Line 371:
Change “figure 2” to “figure 1”
Line 418:
Mention figure 9 here.
Author Response
The review „NMR for Single Ion Magnets” submitted to the journal Magnetochemistry, is written by experts in the field of NMR spectroscopy of paramagnetic molecules. The focus of this article lies on lanthanide based SIMs while touching d-block systems at times. The article covers the theoretical background necessary for understanding the field and gives some examples of NMR studies of SIMs in solution (chapter 3) and in the solid-state (chapter 4). It should be published after a minor revision:
We thank the reviewer for the support and for the constructive indications
Major points of criticism:
- It is clear that polynuclear d-block SMMs like the Mn12cluster are not covered in this article, however I wonder whether NMR studies of dinuclear Ln-based SMMs should be ignored or not ( Am. Chem. Soc. 2013, 135, 14349. DOI: 10.1021/ja4069485 or Nature Communications 2019, 10, 571. DOI: 10.1038/s41467-019-08513-6).
- On the other hand, the first comprehensive NMR study on a derivative of the archetypal SIM TbPc2must definitely be mentioned in such a review (-Eur. J. 2015, 21, 14421. DOI: 10.1002/chem.201501944). Nice NMR studies have also been made with fullerene trapped SIMs (Chemical Science 2015, 6, 2328. DOI: 10.1039/C5SC00154D)
We sincerely apologize for overlooking these studies. We have now duly mentioned them. The sentence about double and triple deckers has now been rewritten as:
A complete NMR characterization of double- and triple-decker (either SIMs with a single paramagnetic center or dinuclear SMMs) has followed more recently, due to the difficulties that are related to the detection of the signals of strongly paramagnetic systems as described above [49,52,76–78]. Of note, porphyrin double deckers were addressed earlier [79,80].
The two papers about caged SMMs have been mentioned as general references.
Minor points:
- Line 34:
I am not aware if there is already an “application” of SIMs. At least the authors do not give a reference. Therefore, I think it would be better to change
“SIMs find large applications in …..”
to “are expected to find large applications in….”
We agree with the reviewer. The text has been changed accordingly.
- Line 141:
- I agree that for several NMR signals of SIMs FCS is negligible, however, it has nevertheless to be considered for some of the resonances (e.g. 13C atoms and aromatic C-1H in LnPc2anion). When the ligands carry a delocalized unpaired electron like in the neutral from of LnPc2 FCS has to be considered. Therefore, I suggest changing from:
“.., except for relatively short metal-nucleus distances.”
to
“.., except for relatively short metal-nucleus distances, or in ligands with considerable electron delocalisation or for ligands carrying unpaired electrons (e.g. neutral double decker phthalocyaninato complexes).”
We thank the reviewer for pointing this out. We have included the sentence with the appropriate reference.
- Chapter 2.1.2 Field Dependent effects
I always appreciated very much when estimations of several effects are presented in numbers or in nice diagrams (e.g. in this article Figures 4 and 5 and many examples in the fantastic book “NMR of paramagnetic molecules”). Therefore it would be great to see an estimation of the field dependant effects presented in equations 10 and 11 (e.g. for TbPc2at a field of 24 Tesla).
This suggestion is very interesting and welcome. We have added an estimate of the relative change in the shift over commercially available fields, in the form of Figure 3.
- Figure 4:
- a) These curves may lead to misunderstandings: Curves are given for S=1/2 and S=7/2. On the other hand it is stated, that electron relaxation depends on J for lanthanides. Therefore the curves presented for S=1/2 or S=7/2 may represent d-block ions? However, S=7/2 is not possible for transition metals. Perhaps the diagrams could be changed to S=5/2 (corresponding to high-spin Mn2+).
We have removed the S=1/2 to clarify the figure and replaced S=7/2 with gadolinium(III), as for gadolinium the relaxation depends on S, as now better specified at line 244 and 277.
- b) the blue dots are hardly visible (especially in the left diagram).
We have tried to make this figure clearer.
- Line 322:
Reference 69 does not seem to be a SIM paper and is inadequate here
This reference has been removed.
- Line 325 - 326:
The statement “… and their NMR characterization has followed shortly after [72,73]” is not correct and needs to be changed. The references given here are not from phthalocyaninato double deckers but from porphyrin double deckers. The NMR characterization of LnPc2-derivatives was done much later.
See response to the first major point above.
- Line 363:
The fact that FCS is negligible here has to do with an “insulating unit –O-B-”. In order to make this clearer to the readers one might insert a sentence like:
“The signals arise from the long alkyl chain attached to the boron atom. As spin density is hardly transmitted through the O-B-unit contact shifts are negligible.”
We thank the reviewer for pointing this out. We have now added: “The 1H-NMR signals arise from the long alkyl chain attached to the boron atom. Because the O-B bonds impede the transmission of the spin density, the contact shifts are negligible.”
- Line 371:
Change “figure 2” to “figure 1”
Done
- Line 418:
Mention figure 9 here.
Done